# The Impact of Fertilizer Gradient on High Nature Value Mountain Grassland

**DOI:** 10.3390/plants14213397

**Published:** 2025-11-06

**Authors:** Costel Samuil, Adrian Ilie Nazare, Culiță Sîrbu, Bogdan Grigoraş, Vasile Vîntu

**Affiliations:** Plant Science Department, Iasi University of Life Sciences, 3, M. Sadoveanu Alley, 700490 Iasi, Romania; costel.samuil@iuls.ro (C.S.); adrian.nazare@iuls.ro (A.I.N.); culita.sirbu@iuls.ro (C.S.); bogdan.grigoras@iuls.ro (B.G.)

**Keywords:** grassland management, input doses, biodiversity, indicator species, Carpathian Mountains, sustainable practices

## Abstract

High nature value (HNV) grasslands in mountain areas are important ecosystems for biodiversity maintenance and offer a multitude of ecosystem services, but they are constantly threatened by abandonment or intensive fertilization. The aim of this study was to assess the effects of organic and mineral fertilization, under mulching and abandonment scenarios, on the floristic composition and diversity of *Nardus stricta*-dominated grasslands located in the North-Eastern Carpathians (Romania). The field experiment included 11 variants (control, low, moderate, and high inputs), analyzed as communities with cluster, ordinations, indicator species, and α indices. The results showed a clear separation of communities along the input gradient, from the oligotrophic grassland dominated by *Nardus stricta* (control variant) to mesotrophic/eutrophic communities dominated by *Dactylis glomerata, Festuca pratensis,* and *Trifolium pratense* at moderate and high inputs. Moderate fertilization (10–20 t ha^−1^ manure; N_50_P_50_K_50_–N_100_P_100_K_100_) maximized species richness (37–38 species), Shannon diversity (H′ = 2.5–2.6), and evenness (E = 0.70–0.75). High inputs reduced diversity and favored competitive grasses. Indicator species analysis highlighted a multitude of species that show the plant communities’ response to adaptive management. Moderate fertilization provides a viable trade-off between productivity and biodiversity, while abandonment or overfertilization accelerates biodiversity loss.

## 1. Introduction

High nature value (HNV) grasslands are mountain ecosystems of strategic importance for biodiversity conservation, providing ecosystem services and maintaining traditional cultural landscapes in Europe [1,2]. They provide essential support, supply, regulation, and cultural services, contributing to carbon sequestration, water retention, fire risk reduction, and the maintenance of natural and cultural heritage [3]. These semi-natural agricultural systems were formed and maintained through extensive management, especially mowing and grazing, and are characterized by high floristic diversity and the presence of indicator species adapted to oligotrophic and stressful conditions [4,5,6]. In the establishment of mountain grasslands management, the soil–plant relationship plays an essential role, with indicator species providing rapid information on the agrochemical conditions of the soil and on the state of fertility [7,8].

The classification and assessment of HNV grasslands is an essential basis for choosing adapted management techniques and for understanding their role in cultural landscapes [9]. The pressures are exacerbated by difficulties in implementing EU conservation policies, due to limited collaboration between environmental and agricultural institutions, leading to an unfavorable conservation status for species-rich semi-natural habitats [10]. Moderate fertilization can stimulate species productivity and coexistence, reducing competitive dominance—a mechanism similar to that suggested by the intermediate disturbance hypothesis [11,12,13,14].

In the last decades, changes in agricultural practices—the intensification of mineral fertilization or the abandonment of traditional management—have led to major transformations in the structure and functioning of mountain grasslands [15,16,17,18]. The abandonment determines secondary successions and major structural changes, with the degradation of agronomic and ecological value [19]. Different studies propose alternative technologies that are economically viable in mountainous areas, such as mulching combined with organic fertilization, which maintain oligotrophic grasslands at an appropriate level of diversity, with minimal changes in vegetation composition [20,21,22]. Several conceptual approaches propose the segmentation of microbial functional niche to assess alterations caused by fertilization, extracting sensitive and stable guilds, which support sustainable management policies in HNV grasslands [23,24]. Another approach can analyze the nitrogen use efficiency (NUE), which is used in the design of sustainable temporary grassland systems, as an indicator for optimizing fertilization and reducing nutrient losses [25].

Romania has some of the largest areas of HNV grasslands in the European Union, especially in the Western and Southern Carpathians, where they amount to over 2 million hectares [6,26,27]. These ecosystems are valued both for their floristic biodiversity and for their cultural and economic value [28,29]. Recent studies in the Apuseni Natural Park (Romania) show that permanent grasslands in habitat 6520 have an average productivity of 5.2 t ha^−1^ green mass and a pastoral value of 35 [30], values that highlight the importance of these ecosystems for maintaining biodiversity and providing fodder resources. Although HNV grasslands support medicinal plants, such as *Arnica montana*, which provide high economic and cultural value, they are constantly threatened by intensive agricultural practices, requiring sustainable conservation strategies [31]. However, maintaining the balance between productivity, biodiversity conservation, and sustainable resource use remains a major challenge for current and future agricultural and environmental policies [32,33]. In this sense, European agricultural policies promote agri-environmental schemes and compensatory measures that support biodiversity, but abandonment and intensification continue to pose significant threats [6,34,35].

In the Apuseni Mountains (Romania), population exodus has accentuated the abandonment of grasslands, leading to the transformation of traditional cultural landscapes and the reduction in biodiversity [36], which highlights the need to restore traditional agricultural practices [37]. Long-term studies conducted in this area have demonstrated that constant application of manure has a significant impact on the floristic composition and productivity of *Festuca rubra* grasslands [38,39]. In contrast, intensive fertilization leads to the simplification of phytocoenoses and the dominance of competitive grasses and perennial legumes, while the lack of input favors oligotrophic species and reduces forage value [40,41,42,43]. Furthermore, long-term fertilization reorients mycorrhizal colonization strategies in the roots of dominant species (e.g., *Agrostis capillaris*), favoring storage structures over transfer ones, which may affect nutrient cycling in HNV ecosystems [44]. The assessment of indicator species for soil agrochemical characterization has also been used in several studies conducted in Apuseni Natural Park [7], providing a useful framework for monitoring and adaptive management of type 6520 habitats. These results can inform conservation policies, where, according to comparative studies, institutional collaboration between conservation and agriculture is essential for maintaining the favorable status of species-rich grasslands, integrating socio-cultural aspects [45].

In this context, the objective of the study was to evaluate the solidity of the hypothesis that different management scenarios will affect the stability of the plant community in a *Nardus stricta* HNV grassland. For this, a multiple-treatment experiment was designed to analyze the effects of organic and mineral fertilization, combined with mulching and abandonment, on the diversity and structure of species. A complex system of data analysis was carried out to identify the most suitable test that answers one or several research questions. (a) Is the application of combined management scenarios visible in the plant communities? (b) Are diversity indices sensitive enough to show the changes due to applied management? (c) Is there a visible shift from an oligotrophic status to a meso- or eutrophicated one? By applying multivariate methods and indicator species analysis, the research aims to highlight the mechanisms by which the intensity and type of input influence the dynamics of mountain phytocoenoses and to provide recommendations for an adaptive and sustainable management of these ecosystems.

## 2. Results

### 2.1. The Influence of the Management Scenario on the Grassland Community

The cluster analysis highlighted the vegetation classification and clearly reflected the ecological gradient induced by the different management practices and fertilization rates (Figure 1). Four distinct clusters were identified: Group 1—oligotrophic grasslands dominated by *Nardus stricta*, typical of nutrient-poor conditions and extensive management; Group 2—transitional communities of *Nardus stricta–Festuca rubra*, reflecting a moderate shift in floristic composition at slightly increased inputs; Group 3—*Festuca rubra–Nardus stricta* communities co-dominant with *Agrostis capillaris*; and Group 4—mesotrophic grasslands of *Agrostis capillaris–Festuca rubra*, associated with high fertilization levels. The formation of these clusters demonstrates that fertilizers produce major changes in the floristic structure of HNV grasslands, separating two large ecological groups: one corresponding to treatments with low or absent input (T1–T3) and another associated with intensive organic or mineral fertilization (T4–T11). Each management regime thus generated a specific type of grassland, highlighting the close relationship between the level of inputs and the composition of the communities.

### 2.2. Plant Community Patterns Explored Through Principal Coordinates Analysis (PCoA)

To more clearly visualize the relationships between treatments and floristic communities, we used PCoA analysis, which highlighted the distribution of phytocoenoses along the trophic gradient generated by fertilization, which is represented by Axis 1 (Figure 2, Table 1).

On this axis, treatments without fertilization (T1–T3) were negatively correlated (*r* = −0.825, *p* < 0.001) and clustered on the left side of the ordinate, while treatments with medium and high inputs (T5–T11) were positively correlated and clustered on the right side. Axis 2 explained only 11.1% of the variation and had low biological relevance. It partially separated variants with medium input from those with high input. Overall, PCoA confirms that fertilization intensity, and not type (organic vs. mineral), explains most of the variation in floristic composition.

The vectors corresponding to the type of fertilizer (organic vs. mineral) had close values and similar directions, indicating the lack of significant differences between the two types of input at the same intensity level. Thus, the variability of the floristic composition is mainly explained by the intensity of fertilization and not by its type.

The Multi-Response Permutation Procedure (MRPP) analysis was applied to test the robustness of group separation based on floristic analysis and to quantify the distance between the identified grassland types. The groups without input (T1–T3) were significantly separated from all fertilized treatments (*p* < 0.01), but the differences between treatments with close doses, either organic or mineral, were not statistically significant (Table 2). The greatest difference was observed between the zero-input group (T1–T3) and the high-input group (T6, T10). Intermediate values of the A index were obtained for the zero-input vs. medium-input and low-input vs. high-input comparisons. The lowest values were recorded between medium input and high input, suggesting a partial convergence of the communities under high fertilization.

### 2.3. The Comparative Analysis of Community Composition and Species Projection Along Fertilization Gradient

To visualize how these correlations are reflected in the community structure, the distribution of species and treatments was graphically represented by PCoA analysis, where the vectors indicate the direction and amplitude of the species response to the fertilization gradient (Figure 3). These differences confirm the clear separation between the grassland groups previously identified by cluster analysis and MRPP tests. The PCoA showed a clear separation between the treatments with low inputs (T1–T3) and those with medium and high levels of mineral and organic fertilization (T4–T11), arranged along Axis 1. This axis explained most of the variation (87.5%) and represented the main trophic gradient. On the negative side of Axis 1 are phytocenoses dominated by *Nardus stricta, Vaccinium myrtillus,* and *Luzula multiflora*, indicating oligotrophic grasslands characteristic of treatments without inputs. On the positive side, medium and high fertilization caused a shift of communities towards competitive mesotrophic species, such as *Dactylis glomerata*, *Festuca pratensis,* and *Phleum pratense*. Axis 2, although explaining only 11.1% of the variation, highlighted the differentiation of leguminous species (*Trifolium pratense*, *Trifolium repens*) associated with moderate organic inputs. This distribution confirms the trends highlighted by the cluster and MRPP, suggesting that the application of inputs not only changes the dominance of species but also the typology of phytocenoses in the long term.

To further explore the contribution of individual species to the separation of phytocenoses highlighted in the PCoA analysis, the correlations between species abundance and the two ordination axes are presented in Table 3.

The coefficient values show which species were favored by the lack of fertilization and which respond positively to moderate or high nutrient inputs. The correlations between ordination scores and the abundance of dominant species were analyzed, showing that species characteristic of oligotrophic grasslands, such as *Nardus stricta*, *Vaccinium myrtillus, Luzula multiflora*, and *Potentilla erecta*, were negatively correlated with Axis 1. This position on the ordination indicates that their ecological optimum is associated with the absence of fertilization or with very low inputs. *Festuca rubra* and *Carex pallescens* also showed the same trend, confirming their vulnerability to management intensification. In contrast, mesotrophic and competitive species had positive and significant correlations with Axis 1, among which *Dactylis glomerata*, *Festuca pratensis*, *Phleum pratense,* and *Agrostis capillaris* stand out (all *p* < 0.001), demonstrating an increase in their abundance under medium and high fertilization conditions. Axis 2, although explaining a smaller proportion of the variation, highlighted a distinct behavior of leguminous species. *Trifolium pratense* and *Trifolium repens* were negatively and significantly correlated with this axis, showing that their response was influenced by the application of moderate organic inputs. Other species, such as *Briza media* and *Veronica chamaedrys*, also presented significant correlations with Axis 2, suggesting the role of secondary ecological factors in structuring communities.

### 2.4. The Indicator Species Analysis (ISA) on Plant Communities Shaped by Management Scenarios

To identify the species responsible for these differences, we used ISA analysis. Analysis of species indicator values for the four phytosociological groups (Table 4) highlighted floristic differentiation along the management and fertilization gradient. Four distinct groups were identified and profiled through this method.

Group 1 (*Nardus stricta* grasslands) is characterized by strong indicators with high significance: *Nardus stricta, Campanula abietina, Cerastium sylvaticum, Cruciata glabra, Polygala vulgaris, Viola declinata, Veronica officinalis,* and *Vaccinium myrtillus*.

Group 2 (transitional communities) is associated with grasses (*Anthoxanthum odoratum, Briza media, Cynosurus cristatus*) and legumes (*Lotus corniculatus, Trifolium ochroleucon, Trifolium repens*).

Group 3 (*Festuca rubra–Agrostis capillaris* co-dominance) is marked by species tolerant to moderate inputs: *Festuca rubra, Holcus lanatus, Trifolium pratense, Alchemilla vulgaris, Centaurea pseudophrygia,* and *Stellaria graminea*.

Group 4 (competitive mesotrophs) has indicators such as *Agrostis capillaris, Dactylis glomerata, Festuca pratensis, Phleum pratense* but also generalist species (*Taraxacum officinale, Veronica chamaedrys*).

Thus, the indicator species analysis (ISA) confirmed that each phytosociological group has a distinct set of characteristic species, which validates the separation of clusters and the ordination gradient obtained by PCoA.

### 2.5. The Impact of Management Scenarios on Diversity Indices

To evaluate the effects of fertilization on α diversity, four biodiversity indices (S, H′, E, D) were calculated, and the results are presented in Table 5. The overall assessment of biodiversity indices highlights a significant effect of fertilization treatments on the structure of phytocenoses.

Species richness (S) had the highest values in the variants with moderate inputs (T4—37 spp., T7—38 spp., T9—37 spp.), confirming the positive effect of moderate organic or mineral fertilization on diversity. The lowest values were recorded in the treatments with high doses of fertilizers (T6—22.5 spp., T10—22.8 spp.), where the competitive pressure of dominant species led to a reduction in the total number of species.

The Shannon index (H′) increased significantly in the fertilized treatments (T4–T9, values > 2.4) compared to the control (T1—1.54) and the abandonment variant (T2—1.30). This shows a more balanced distribution of species and a higher diversity under moderate fertilization conditions. In contrast, in very high doses of inputs (T6, T10), although the Shannon index remains relatively high (>2.1), diversity tends to be supported by a few dominant species.

Species evenness (E) followed the same trend: low values in the control and abandonment (0.43 and 0.38) compared to the highest values in the fertilized treatments (0.70–0.75). This indicates that fertilization favors a more uniform distribution of species abundance, up to an input threshold, after which the communities become unbalanced again.

The Simpson index (D) reflected the same dynamics, with higher values in fertilized treatments (0.83–0.86) compared to the control and abandonment (0.51, 0.42, respectively).

It is observed that moderate fertilization (T4, T7, T9) maximizes α diversity, which confirms the hypothesis that intermediate inputs favor species coexistence by reducing competitive imbalances. These results support the intermediate disturbance hypothesis, indicating that moderate fertilization maximizes species coexistence, while both the absence of inputs and excessive fertilization lead to a reduction in diversity.

## 3. Discussion

### 3.1. The Changes of Different Management Scenarios in the Assemblage of Grassland Communities

The long-term application of fertilizer on grasslands represents a disturbance for plant communities, which is visible in the changes of species richness, shifts between dominant species, and community simplification. All these changes can be used as predictors in the intermediate disturbance hypotheses and provide predictability of succession and change. The application of organic and mineral fertilizers determined a clear difference in the evolution of phytocoenoses, confirming the hypothesis that management intensity is the main factor modulating the structure and diversity of grasslands [46,47]. In unfertilized plots (T1), the persistence of a diverse plant community dominated by *Nardus stricta* indicates that nutrient limitation acts as a stabilizing ecological filter, maintaining oligotrophic conditions and preventing the dominance of competitive species. This supports the view that low-input systems preserve high functional redundancy, ensuring ecological resilience under fluctuating climatic conditions. Comparable findings from mountain grasslands in Romania suggest that the maintenance of traditional management or periodic mowing is essential to sustain this equilibrium. [48].

The decrease in species richness and evenness observed under abandonment (T2) suggests that the cessation of management triggers directional succession toward ruderal and nitrophilous assemblages. In the absence of disturbance, nutrient accumulation and canopy closure promote the dominance of tall grasses and competitive forbs, suppressing stress-tolerant oligotrophic species. These dynamics highlight that complete abandonment, although often perceived as beneficial for conservation, can paradoxically reduce biodiversity in high-altitude grasslands by disrupting the disturbance–productivity balance that maintains species coexistence. [49]. The reduction in specific richness observed in the case of abandonment was previously reported by [19], who showed that after 6–10 years of non-use, diversity and forage value decrease, being replaced by low-value species, such as *Nardus stricta*. In contrast, mulched variants (T3) maintained high diversity, indicating that this type of extensive management can substitute traditional mowing and prevent the loss of valuable species [21,22].

Mineral fertilization (T4–T6) induced contrasting ecological responses along the fertilizer gradient. At moderate doses, the balance between nutrient enrichment and competitive pressure favored the coexistence of stress-tolerant and fast-growing species, resulting in the highest levels of diversity. This response illustrates the intermediate disturbance hypothesis, where moderate disturbance intensity maintains coexistence by preventing the dominance of a few competitive taxa and sustaining structural heterogeneity [11,50]. Conversely, excessive nutrient inputs triggered a shift toward communities dominated by tall nitrophilous grasses and competitive legumes, reducing species richness and evenness. Such patterns, also reported for other high nature value grasslands, suggest that high fertilization disrupts resource partitioning and light availability, leading to competitive exclusion and loss of ecological resilience. [20,41,42,51,52]. Similar studies from the Apuseni Mountains (Romania), using NIR techniques, confirm the decrease in protein content and digestibility under intensive NPK, linked to the reduction in Fabaceae, and recommend moderate doses or organic fertilization to maintain biodiversity and forage quality [53], which highlights the importance of integrated management in HNV.

Moderate organic fertilization (10–20 t ha^−1^ of cattle manure) promoted the recovery of perennial legumes, such as *Trifolium pratense* and *Lotus corniculatus*, enhancing both floristic diversity and forage quality. This response reflects the dual ecological function of organic amendments, which gradually release nutrients and improve soil microbial activity without disrupting the oligotrophic balance of mountain grasslands. The slow nutrient mineralization and increase in soil organic matter create favorable microsites for legume establishment, supporting nitrogen fixation and stabilizing the plant community. Similar effects have been reported in other mountain ecosystems, where moderate organic inputs sustain long-term productivity while preserving species coexistence [6,54]. Our results are comparable to long-term studies carried out on *Festuca rubra* meadows in the Apuseni Mountains, where it was observed that the dose of 10 t ha^−1^ contributed to balancing the floristic composition and increasing the Shannon index, suggesting that moderate organic fertilization can support both productivity and maintain diversity [38]. At the same time, periodic fertilization with 20 t ha^−1^ every two years (T9) allowed maintaining a balance between productivity and biodiversity, the results being similar to those reported by [6] and [55] for HNV grasslands in the Romanian Carpathians.

At high manure doses (T10, T11), diversity was again restricted, with values close to those obtained for intensive mineral fertilization. These results indicate that both zero and excessive inputs lead to biodiversity loss, while moderate doses provide a good compromise between productivity and species conservation [28,56,57]. Our results confirm that low NPK doses do not produce major structural changes, in line with 6-year studies in the Apuseni Mountains, where it was shown that low fertilization does not change grassland type but influences species diversity [58].

The nonlinear response of diversity to the fertilization gradient can be explained by ecological mechanisms related to asymmetric competition for light and nutrients. At moderate doses, increased resource availability reduces the advantage of dominant species and allows stress-tolerant species to coexist, which maximizes diversity. In contrast, at high inputs, excessive fertilization favors nitrophilic species and competitive grasses, which, by increasing biomass and shading the lower layer, gradually exclude oligotrophic species adapted to poor soils. This transition from balanced communities to those dominated by a few opportunistic species explains the sharp decline in Shannon and Simpson indices at high fertilization rates.

### 3.2. Plant Specificity to Applied Treatments

The results of ISA show that each identified phytocoenotic group has a distinct set of species with diagnostic value. Oligotrophic grasslands are characterized *by Nardus stricta, Vaccinium myrtillus,* and *Potentilla erecta*, species typical for acidic soils and restrictive conditions [5], which indicate a native adaptation to reduced nutrient conditions. Several recent studies confirm that *Nardus stricta* is a characteristic species for oligotrophic grasslands on acidic soils, such as the case of reserves in Germany, where typical acidophilic species decrease with the improvement of soil acidity [59]. The presence and the coverage of this species can be used to indicate the shifts from acidic to neutral or alkaline conditions. *Vaccinium myrtillus* also thrives in restricted acidophilic environments, such as degraded grasslands [60,61]. The case study of *Nardus stricta* grasslands (6230) provides further evidence for the role of these species as ecological indicators in oligotrophic communities [8,62]. In contrast to oligotrophic grasslands, mesotrophic and eutrophic communities were dominated by species with higher nutritional requirements, such as *Dactylis glomerata, Festuca pratensis,* and *Trifolium pratense*. These species are recognized as competitive under moderate and intense fertilization conditions, where biomass accumulation and the reduction in species sensitive to low nutrient inputs are favored. This opposition between species characteristic of oligotrophic environments and those associated with mesotrophic/eutrophic communities highlights the impact of the fertilization gradient on the structure of mountain vegetation. Similar results have been reported in recent studies on mixtures of grasses and legumes fertilized with mineral and organic fertilizers, where *Dactylis glomerata* and *Trifolium pratense* became dominant [63,64]. Parallel findings have also been presented for high nature value mountain grasslands in the Apuseni Mountains, where increased fertilization levels have led to an increased role for mesotrophic and eutrophic species [6]. Such results confirm the role of indicator species as essential tools for diagnosing the agrochemical status of the soil and for underpinning adaptive management strategies for mountain grasslands [7,8,40].

Overall, the results demonstrate that *Nardus stricta* meadows respond differently to the intensity and type of fertilization, but the general trend remains the same: extensive or moderate management preserves diversity, while excessive intensification leads to simplification of the phytocoenosis. These findings support the recommendations for adaptive management and sustainable use of mountain resources [59,65,66,67]. Changes generated by fertilization are subsequently amplified or attenuated by climate variability, which complicates predictions of the long-term dynamics of these ecosystems. Although fertilization treatments explain a large part of the observed variation in vegetation composition, these changes cannot be interpreted as isolated cases. During the experimental period (2020–2024), the average annual temperature in the study area ranged between 5.9 and 6.8 °C, and the total amount of precipitation between 620 and 699 mm, with a pronounced minimum in 2024. These moderate but significant climatic variations in the mountain context may partly explain the annual fluctuations in diversity indices, particularly the slight reduction observed in the last year, when the precipitation deficit accentuated water stress and interspecific competition. Thus, the effects of fertilization on community structure cannot be analyzed in isolation but in correlation with the thermal and water regime, which modulates the response of oligotrophic and mesotrophic species in mountain conditions. Interannual climatic fluctuations (temperature, water balance) can amplify or attenuate the effects of management, generating both directional successions and annual variations in the structure of phytocoenoses. This aspect is particularly relevant for mountain grasslands, where the interaction between anthropogenic and climatic factors determines the dynamics of plant communities. Studies in the Apuseni Mountains show that these changes are magnified by climatic fluctuations, which induce directional successions in initial periods and annual fluctuations related to temperature and water balance, affecting the vegetation composition in similar grasslands [68]. Climate fluctuations can increase the speed of vegetation changes in mountain grasslands—different studies show directional successions towards species adapted to warmer and drier conditions [69] and changes in floral composition under warming conditions [70]. These results indicate a background of a high global sensitivity of vegetation indicators to temperature and precipitation (global meta-analysis) and dramatic examples, such as the effects of the extreme drought of 2022 [71]. The context demands the necessity to integrate climate monitoring into HNV management in order to distinguish between human-induced and natural changes—as demonstrated by integrated monitoring programs for HNV [72], the German standardized observation system [73], and the international GLORIA network dedicated to alpine eco-vegetation [74].

### 3.3. Species Structure and Diversity Under the Input Gradients

The results of our study demonstrate that fertilization intensity plays a central role in shaping the structure and diversity of high nature value (HNV) grasslands. There is a clear separation of grassland communities along the input gradient, from unfertilized oligotrophic grasslands to intensively fertilized mesotrophic systems. This pattern highlights the sensitivity of oligotrophic grasslands, dominated by *Nardus stricta* and associated stress-tolerant species, to increased nutrient availability. Also, intensive fertilization reduces diversity and favors competitive species, potentially increasing the risk of toxic species in grasslands [75]. To prevent animal poisoning, technologies such as NIR hyperspectral imaging can discriminate species and botanical families, allowing the identification of toxic species in grasslands similar to those in the Apuseni Mountains [76]. This highlights the need for advanced monitoring in sustainable HNV management. Our results are consistent with previous reports showing that moderate fertilization can improve species coexistence by balancing competitive dynamics, while both the absence of inputs and excessive fertilization reduce biodiversity [6,46,57,77,78]. Intermediate input treatments (T4, T7, T9) maximized species richness and Shannon diversity, supporting the intermediate disturbance hypothesis in grassland ecosystems [11,12,50].

Overall, the results highlight that reduced or absent inputs favor oligotrophic and stress-tolerant species [5], thus maintaining the traditional character and high biodiversity of mountain meadows. In contrast, intense fertilization causes a transition towards communities dominated by competitive grasses and perennial legumes, with a visible reduction in floristic diversity. These findings are in agreement with both national studies [6,40,46,79] and recent international research showing that excessive mineral fertilization increases biomass but significantly reduces species richness and community evenness [47,57]. Also, the global analysis carried out by [69] confirms that climatic fluctuations can amplify these effects, accelerating the succession towards simpler mesotrophic communities and reducing the stability of ecosystems. Thus, the transition from oligotrophic to mesotrophic systems, accompanied by gradual losses of diversity, reflects a general pattern recently reported for mountain ecosystems in Europe [26,33,70].

For high natural value grasslands, the application of moderate fertilization represents an optimal solution for maintaining the balance between productivity and biodiversity and sustainable long-term management. Extreme management and high-input fertilization or abandonment, the marginal effects of the total lack of inputs to excessive fertilization, lead to the simplification of plant communities. Both these extremes have a great magnitude impact and lead to biodiversity losses and dominance shifting and produce a high instability of community assemblage. This opposition between the species characteristic of oligotrophic environments (*Nardus stricta, Vaccinium myrtillus, Potentilla erecta*) and those associated with mesotrophic and eutrophic communities (*Dactylis glomerata, Festuca pratensis, Trifolium pratense*) highlights the importance of the fertilization gradient as a determining factor in the structure and dynamics of mountain grasslands. Maintaining the balance between productivity and diversity depends on adaptive management, which integrates both agricultural pressure and climatic variability, which is the only viable option for the conservation of biodiversity and the sustainable use of resources in the mountain area.

The study demonstrates that fertilization intensity is the main determinant of the structure and diversity of high nature value (HNV) grasslands. Potential limitations of this are associated with the short period of experimentation (only 4 years), which offers a a good image of the potential change trends. The continuity in experimentation will offer a complex image of both the successional trajectories of vegetation and the resilience potential of plant communities. The study area limits the application of the results to similar types of grasslands, but it needs to be replicated in other climatic conditions to assess the combined and unique impact of treatments and climate. Interdisciplinarity is a good solution to connect the changes in plant communities and species dynamics with soil characteristics and evolution due to the application of treatments.

At the applied level, a clear definition of oligotrophic grasslands is essential for guiding sustainable management strategies and for maintaining the HNV character of these ecosystems. The results show that moderate input can represent a viable compromise between productivity and biodiversity conservation, maintaining the HNV character of grasslands. The identification of species with an indicator value for each level of intensification provides a useful tool for monitoring changes and implementing adaptive management strategies.

## 4. Materials and Methods

### 4.1. Soil and Climatic Conditions

The experiment was conducted during the 2020–2024 period at a permanent grassland derived from *Nardus stricta* L. on the Plaiu Sarului (47030’42’’ N, 25037’95’’ E) in the region of the North-Eastern Carpathians (Dorna Depression) at 845 m above sea level. The type of soil within the experimental field is represented by luvisol, in which a well-developed thatch of dead and living stems, leaves, and roots with a thickness of 8–10 cm is noticeable [80]. Mineral soil horizon of moderate humus accumulation (2.43%) below thatch layer is light in color with moderately acidic reaction (pH = 5.4) and medium texture (content of clay 22,3%). The eluvial horizon, 42 cm thick, is weakly skeletal with medium texture (clay = 19.5%) and is moderately acidic (pH = 5.1). The value of the cation-exchange capacity from the humiferous horizon (Ao) is 11.7 meq 100^−1^ g of soil, and the degree of base saturation is 64,3%. The humus accumulation horizon is moderately supplied with nitrogen (0.952%) and potassium (257 ppm) and poorly supplied with mobile phosphorus (12.4 ppm).

The climatic conditions in the area are characterized by an average temperature of 6.3 °C and 675 mm of total annual precipitation. In the Plaiu Sarului area, the amounts of atmospheric precipitation are characteristic of depressional mountain areas. Thus, the months with the lowest precipitation are January, February, March, and December (32.4 mm to 38.9 mm), and the months with the highest precipitation are obviously the summer months of May, June, July, and August (85.4 mm to 118.2 mm). The relative humidity of the air has high values throughout the year of 80–92%, with an average value of 85.2%. The precipitation recorded during the experimental period was 698.9 mm in 2020, 677.2 mm in 2021, 695.5 mm in 2022, 680.1 mm in 2023, and 620.6 mm in 2024. The average annual temperatures were 6.8 °C in 2020, 5.9 °C in 2021, 6.3 °C in 2022, 6.6 °C in 2023, and 6.7 °C in 2024.

### 4.2. Experimental Design

A monofactorial experiment with 11 treatments was organized according to the randomized block method in four replicates.

This classification reflects the reference to the thresholds established by the Nitrates Directive (170 kg N ha^−1^year^−1^), as well as the ecological response of the vegetation observed in the field [81, 82].

The research covered four blocks, and the experiment had the following treatments:•T1—unfertilized (control);•T 2—abandonment (unharvested or non-grazing);•T3—mulching (cut and leave the biomass on site);•T4—fertilization with 50 kg N ha^−1^, 50 kg P_2_O_5_ ha^−1^, and 50 kg K_2_O ha^−1^ applied annually;•T5—fertilization with 100 kg N ha^−1^, 100 kg P_2_O_5_ ha^−1^, and 100 kg K_2_O ha^−1^ annually;•T6—fertilization with 150 kg N ha^−1^, 150 kg P_2_O_5_ ha^−1^, and 150 kg K_2_O ha^−1^ annually;•T7—10 t ha^−1^ cattle manure annually;•T8—20 t ha^−1^ cattle manure annually;•T9—20 t ha^−1^ cattle manure at 2 years;•T10—30 t ha^−1^ cattle manure annually;•T11—30 t ha^−1^ cattle manure at 2 years.

Two types of fertilizers were used: an organic one represented by well-fermented cattle manure (older than two years) and a mineral one represented by a complex fertilizer with nitrogen, phosphorus, and potassium at a ratio of 20:20:20 (N:P_2_O_5_:K_2_O). The cattle manure had 16% dry matter and the following chemical composition: N—0.445%, P_2_O_5_—0.212%, and K_2_O—0.695%. Fertilizers were applied manually in the same period, in early spring, before the start of active vegetation growth, respectively.

### 4.3. Vegetation Survey

The floristic studies were performed according to a modified Braun–Blanquét method (Figure 4, Table 6), with smaller intervals adapted for a better and more realistic vegetation assessment [81]. Floristic studies were performed in 2024, at the beginning of July, when the grasses were in the phenological flowering phase. The optimal time for determining the floristic composition is in July, when the grasses are in bloom, which ensures the correctness of the composition of the floristic survey, as shown by other studies conducted on semi-natural grasslands [82,83]. Mowing was performed in the experimental field once per year at a height of 4 cm in the middle of July.

### 4.4. Data Analysis

In order to completely explore the data and to analyze the qualitative effect of treatments, we have used four categories of treatments: zero input, low input, medium input, and high input*.* This approach can be used to additionally quantify the effect of fertilizer as reported by the applied quantity. To integrate the experimental treatments (T1–T11) into the multivariate analyses, they were coded according to the level of nutrient inputs and the type of fertilizer: “Zero-input” category, characterized by the lack of nutrient input (T1–T3); low-inpu*t* category: T4 and T7; medium-input category: T5, T8, T9, and T11; and high-input category: T6 and T10. The differentiation between mineral and organic fertilization was introduced by additional binary variables (e.g., Low_o, Low_m, Med_o, Med_m), allowing the separate testing of the effects of the type of fertilizer on the community structure. The procedure for classifying the experimental treatments by input categories (“Zero-input”, “Low-input”, “Medium-input”, “High-input”) follows common practices in the experimental ecology literature, where treatments with different levels of fertilization (or nutrient input) are compared for effects on plant diversity and productivity [84,85].

Data analysis was performed with PC-ORD 7 software [86,87]. For cluster analysis, the Sørensen similarity index (Bray–Curtis) and the UPGMA (group average linkage) method were used. Both the index and the method were applied due to their ability to show a clear delimitation between ecological and phytosociological groups from the analyzed plant communities [88,89,90]. The cutoff level of the dendrogram was set at approximately 60% of the remaining information, which allowed obtaining clusters with ecological and phytosociological relevance. The choice of this threshold aligns with the methodologies used in recent analyses, where UPGMA cutoff values based on Sørensen/Bray–Curtis were used to obtain ecologically meaningful groupings [14,90].

Ordination analysis was performed by PCoA (Principal Coordinates Analysis) using the Sørensen (Bray–Curtis) distance, a method frequently applied in ecology to compare plant communities based on floristic composition. The choice of PCoA was based on its advantages over iterative methods, such as NMDS, as it provides a unique and reproducible solution, with direct interpretation of distances between samples. In addition, the Bray–Curtis distance is considered one of the most robust dissimilarity measures for vegetation data, being widely used in recent studies of fertilization and management of grasslands and soils [91,92,93]. For each experimental variant, vectors corresponding to fertilization levels (zero, low, medium, high) and input type (organic or mineral) were drawn. The vectors were normalized, and their significance was tested by permutations (n = 999). PC-ORD offers ordination options based on Bray–Curtis distances, overlays or joint plots, and randomization tests, which makes the choice to use vectors corresponding to fertilization levels consistent with good methodological practices [87,94]. The first two ordination axes were retained for interpretation, as they together explained over 95% of the variation in floristic composition.

To test the differences between the groups identified by cluster and ordination analysis, MRPP (Multi-Response Permutation Procedure) analysis was applied. Statistical significance was assessed at the α = 0.05 level. Additionally, indicator species analysis (ISA) [95] was used to identify the species characteristic of each grassland group. For each experimental treatment, α diversity indices were calculated—species richness (S), Shannon–Wiener index (H′), species evenness (E), and Simpson index (D). These indices are widely used in grassland ecology studies, being considered robust tools for assessing the diversity and balance of plant communities [51,96,97,98]. The formulas for these indices are standard in community ecology [99] and were calculated based on relative abundance data. Differences between treatments were tested by one-way ANOVA using SPSS version 29 Academic and scored for significant differences by the LSD test (*p* < 0.05).

## 5. Conclusions

A clear separation of communities along the input gradient, from oligotrophic grasslands dominated by stress-tolerant species to mesotrophic systems characterized by competitive grasses, was recorded.

Moderate fertilization maximizes species richness and floristic diversity by maintaining a balance between competing species and those adapted to restrictive conditions.

Both the abandonment and the application of excessive input cause a reduction in biodiversity, favoring the dominance of a few tolerant or competitive species.

Each management scenario presents a list of indicator species associated with oligotrophic, mesotrophic, and eutrophic communities.

The controlled application of organic and mineral fertilizers, below critical thresholds, emerges as an effective solution for the sustainable use of mountain plant resources.

## Figures and Tables

**Figure 1 plants-14-03397-f001:**
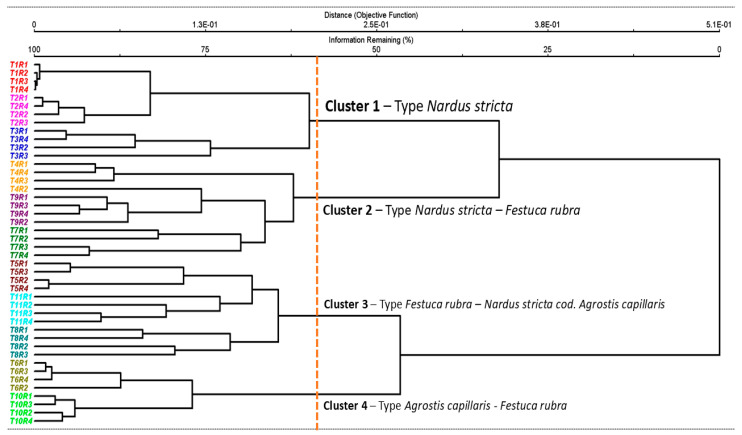
Floristic classification of the vegetation and the changes of grassland type based on cluster analysis. Legend: T1—unfertilized control; T2—abandonment; T3—mulching; T4—50 kg N ha^−1^, 50 kg P_2_O_5_ ha^−1^, and 50 kg K_2_O ha^−1^ mineral fertilization; T5—100 kg N ha^−1^, 100 kg P_2_O_5_ ha^−1^, and 100 kg K_2_O ha^−1^ mineral fertilization; T6—150 kg N ha^−1^, 150 kg P_2_O_5_ ha^−1^, and 150 kg K_2_O ha^−1^ mineral fertilization; T7—10 t ha^−1^ cattle manure applied annually; T8—20 t ha^−1^ cattle manure applied annually; T9—20 t ha^−1^ cattle manure applied every two years; T10—30 t ha^−1^ cattle manure applied annually; T11—30 t ha^−1^ cattle manure applied every two years; R1–R4—replications. The dendrogram cutoff line was set at ~60% remaining information, resulting in four ecologically interpretable clusters.

**Figure 2 plants-14-03397-f002:**
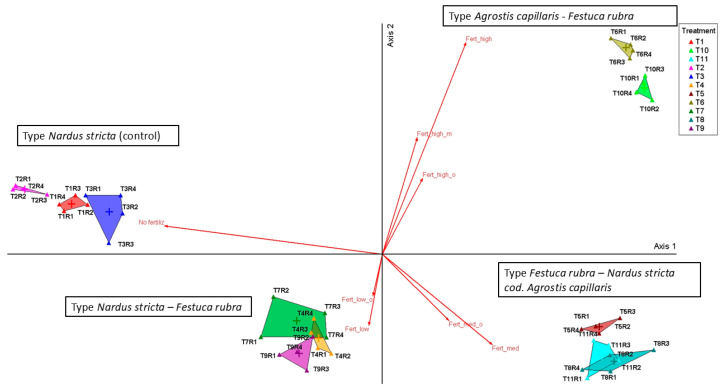
Principal Coordinates Analysis (PCoA) of grassland communities under different fertilization regimes. Legend: T1—unfertilized control (martor); T2—abandonment; T3—mulching; T4—50 kg N ha^−1^, 50 kg P_2_O_5_ ha^−1^, and 50 kg K_2_O ha^−1^ mineral fertilization; T5—100 kg N ha^−1^, 100 kg P_2_O_5_ ha^−1^, and 100 kg K_2_O ha^−1^ mineral fertilization; T6—150 kg N ha^−1^, 150 kg P_2_O_5_ ha^−1^, and 150 kg K_2_O ha^−1^ mineral fertilization; T7—10 t ha^−1^ cattle manure applied annually; T8—20 t ha^−1^ cattle manure applied annually; T9—20 t ha^−1^ cattle manure applied every two years; T10—30 t ha^−1^ cattle manure applied annually; T11—30 t ha^−1^ cattle manure applied every two years; R1–R4—replications. Vectors indicate the fertilization gradient: zero input (no fertilizers applied: T1–T3), low input (reduced doses: T4, T7), medium input (moderate doses: T5, T8, T9, T11), and high input (intensive fertilization: T6, T10), with separation between organic (o) and mineral (m) treatments.

**Figure 3 plants-14-03397-f003:**
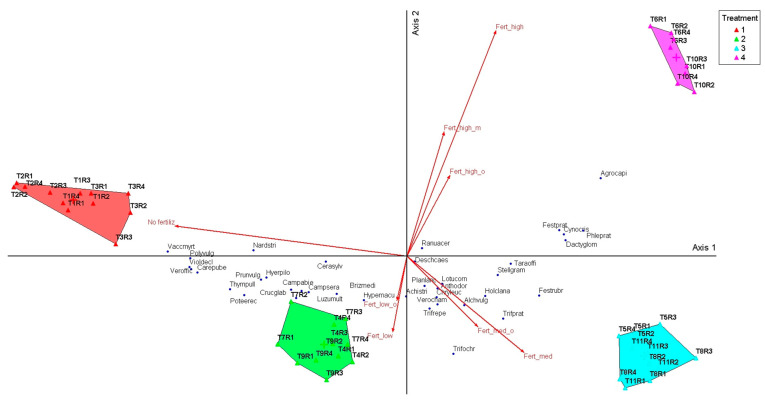
Principal Coordinates Analysis (PCoA) of floristic composition in relation to intensity of treatments. Legend: No fertiliz—no fertilization; *T1*—unfertilized control; *T2*—abandonment; No fertiliz *T3*—mulching; Fert_low *T4*—50 kg N ha^−1^, 50 kg P_2_O_5_ ha^−1^, and 50 kg K_2_O ha^−1^ mineral fertilization; Fert_medi *T5*—100 kg N ha^−1^, 100 kg P_2_O_5_ ha^−1^, and 100 kg K_2_O ha^−1^ mineral fertilization; Fert_hight T6—150 kg N ha^−1^, 150 kg P_2_O_5_ ha^−1^, and 150 kg K_2_O ha^−1^ mineral fertilization; Fert_low_o T7—10 t ha^−1^ cattle manure annually; Fert_medi_o *T8*—20 t ha^−1^ cattle manure annually; Fert_medi_o *T9*—20 t ha^−1^ cattle manure applied every two years; Fert_hight_o T10—30 t ha^−1^ cattle manure annually; Fert_medi_o T11—30 t ha^−1^ cattle manure applied every two years. Species abbreviations: Agrocapi*—Agrostis capillaris* L.; Alchvulg*—Alchemilla vulgaris* L.; Achidist*—Achillea distants *Waldst. & Kit. ex *Willd*.; Anthodor*—Anthoxanthum odoratum* L.; Brizmedi*—Briza media* L., Campabie*—Campanula abietina *Griseb. & Schenk.; Campsera*-Campanula serata *Kit. ex Schult.; Carepube*—Carex pubescens* L.; Cerasyiv*—Cerastium sylvaticum* L.; Cynocysi*—Cynosurus cristatus* L.; Crucglab*- Cruciata glabra* L.; Chryleuc*—Leucanthemum vulgare* Lam., Dactglom*—Dactylis glomerata* L.; Deschcaes*—Deschampsia caespitosa* L.; Festprat*—Festuca pratensis *Huds.; Festrubr*—Festuca rubra* L.; Hierpilo*—Hieracium pilosela* L.; Hypemacu*—Hypericum maculatum *Crantz; Holclana*—Holcus lanatus* L.; Luzumult*—Luzula multiflora *(Ehrh.) Lej.; Lotuscor*—Lotus corniculatus* L.; Nardstri-*Nardus stricta* L.; Phleumprat*—Phleum pratense* L.; Planlanc*—Plantago lanceolata* L.; Prunvulg.*—Prunella vulgaris* L.; Poteerec*—Potentilla erecta* L.; Polyvulg*—Polygala vulgaris* L.; Poterect*—Potentilla erecta (L.) Raeusch.; *Ranuacer*—Ranunculus acer* L.; Stellgram*—Stellaria graminea* L.; Taraoffi*—Taraxacum officinale *Weber ex F.H.Wigg.; Trifprat*—Trifolium pratense* L.; Trifrepe*—Trifolium repens* L.; Trifochr*—Trifolium ochroleucon* L.; Thympule*—Thymus pulegiodes* L.; Verooffi*—Veronica officinalis* L.; Verocham*—Veronica chamaedrys* L.; Vaccmyrt*—Vaccinium myrtillus* L.; Violdecl*—Viola declinata* L.

**Figure 4 plants-14-03397-f004:**
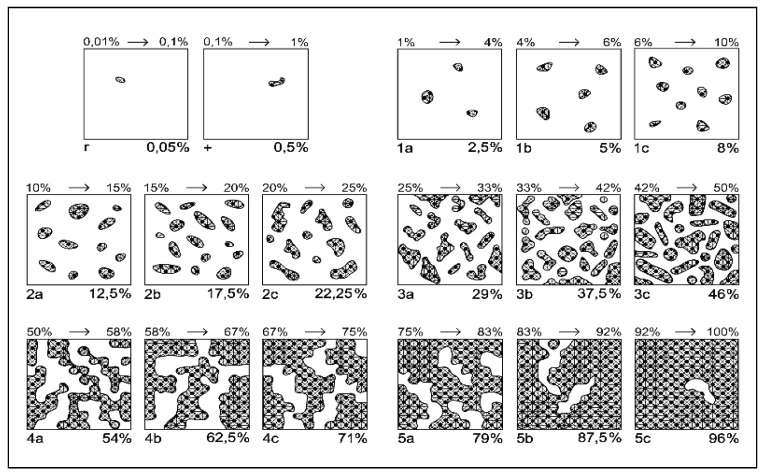
Modified Braun–Blanquét scale for grasslands, based on species [81]. Legend: 1 to 5 indicates the class of coverage; a,b,c indicate the sub-note of each class.

**Table 1 plants-14-03397-t001:** Correlation of experimental factors with the ordination axes (PCoA).

Experimental Factors	Axis 1 (*r*)	Sig.	Axis 2 (*r*)	Sig.
No fertilization (T1–T3)	−0.825	***	0.224	ns
Fertilization—low (T4, T7)	0.352	*	−0.315	ns
Fertilization—medium (T5, T8, T9, T11)	0.586	**	0.174	ns
Fertilization—high (T6, T10)	0.512	**	−0.241	ns
Organic inputs (all levels)	0.468	**	0.112	ns
Mineral inputs (all levels)	0.455	**	−0.098	ns
Axis importance	87.5%		11.1%	ns

Note: *r*—correlation coefficient between ordination distances and explanatory variables; significance: *** *p* < 0.001, ** *p* < 0.01, and * *p* < 0.05; ns = not significant. Axis importance indicates the proportion of total variance explained by each axis.

**Table 2 plants-14-03397-t002:** MRPP pairwise comparisons of floristic composition between the four grassland groups.

Groups Compared	T Statistic	A (Within-Group Agreement)	*p*-Value
Group 1 vs. Group 2	−15.43	0.513	<0.001
Group 1 vs. Group 3	−15.75	0.689	<0.001
Group 1 vs. Group 4	−12.95	0.723	<0.001
Group 2 vs. Group 3	−15.69	0.535	<0.001
Group 2 vs. Group 4	−12.95	0.623	<0.001
Group 3 vs. Group 4	−12.87	0.519	<0.001

Note: T = test statistic; A = chance-corrected within-group agreement. Significance: *p*= zero input (T1: unfertilized control, T2: abandonment, T3: mulching); Group 2 = low input (T4—50 kg N ha^−1^, 50 kg P_2_O_5_ ha^−1^, and 50 kg K_2_O ha^−1^ mineral fertilization, T7: 10 t ha^−1^ manure annually, T9: 20 t ha^−1^ manure every two years); Group 3 = medium input (T5: 100 kg N ha^−1^, 100 kg P_2_O_5_ ha^−1^, and 100 kg K_2_O ha^−1^ mineral fertilization, T8: 20 t ha^−1^ manure annually, T11: 30 t ha^−1^ manure every two years); Group 4 = high input (T6: 150 kg N ha^−1^, 150 kg P_2_O_5_ ha^−1^, and 150 kg K_2_O ha^−1^ mineral fertilization;T10: 30 t ha^−1^ manure annually).

**Table 3 plants-14-03397-t003:** Species projection on PCoA and proximity to fertilization gradient.

Species	Axis 1 (*r*)	Axis 1 (*r*-sq)	Axis 1 (tau)	Signif.	Axis 2 (r)	Axis 2 (*r*-sq)	Axis 2 (tau)	Signif.
*Agrostis capillaris*	0.735	0.540	0.788	***	0.635	0.403	−0.006	ns
*Anthoxanthum odoratum*	0.163	0.026	0.171	ns	−0.368	0.136	−0.270	*
*Briza media*	−0.423	0.179	−0.410	*	−0.488	0.238	−0.319	**
*Cynosurus cristatus*	0.761	0.579	0.699	***	0.227	0.052	−0.127	ns
*Dactylis glomerata*	0.859	0.737	0.646	***	0.186	0.035	0.053	ns
*Festuca pratensis*	0.819	0.670	0.732	***	0.301	0.090	0.043	ns
*Festuca rubra*	0.768	0.591	0.567	***	−0.494	0.244	−0.309	*
*Nardus stricta*	−0.987	0.974	−0.906	***	0.083	0.007	0.107	ns
*Phleum pratense*	0.845	0.714	0.754	***	0.262	0.069	−0.050	ns
*Trifolium pratense*	0.544	0.296	0.356	**	−0.762	0.581	−0.565	***
*Trifolium repens*	0.144	0.021	0.095	ns	−0.719	0.517	−0.522	***
*Luzula multiflora*	−0.525	0.275	−0.451	**	−0.426	0.181	−0.287	*
*Potentilla erecta*	−0.634	0.402	−0.506	***	−0.328	0.108	−0.176	ns
*Veronica chamaedrys*	0.137	0.019	0.093	ns	−0.465	0.216	−0.433	*
*Vaccinium myrtillus*	−0.848	0.719	−0.654	***	0.037	0.001	0.170	ns
*Carex pallescens*	−0.452	0.204	−0.387	*	−0.312	0.097	−0.243	ns
*Ranunculus acris*	0.236	0.056	0.139	ns	−0.268	0.072	−0.201	ns
*Rumex acetosella*	0.718	0.515	−0.535	***	0.059	0.003	0.010	ns
*Stellaria graminea*	−0.023	0.001	0.025	ns	−0.240	0.057	0.205	ns
*Taraxacum officinale*	0.702	0.493	−0.554	***	−0.258	0.066	0.194	ns

Note: *r*—Pearson correlation coefficient; *r*-sq—coefficient of determination; tau—Kendall’s tau correlation between ordination scores and species abundances. Significance: *** *p* < 0.001, ** *p* < 0.01, and * *p* < 0.05; ns = not significant.

**Table 4 plants-14-03397-t004:** Indicator value of plant species under applied treatments.

Species	Group	IndVal	Signif.
*Agrostis capillaris*	4	66.7	*p* < 0.001
*Anthoxanthum odoratum*	2	50.2	*p* < 0.001
*Briza media*	2	45.2	*p* < 0.001
*Cynosurus cristatus*	4	46.6	*p* < 0.001
*Dactylis glomerata*	4	42.4	*p* < 0.001
*Deschampsia caespitosa*	3	27.8	ns
*Festuca pratensis*	4	46.7	*p* < 0.001
*Festuca rubra*	3	52.3	*p* < 0.001
*Holcus lanatus*	3	54.9	*p* < 0.001
*Nardus stricta*	1	50.4	*p* < 0.001
*Phleum pratense*	4	48.1	*p* < 0.001
*Lotus corniculatus*	2	36.2	*p* < 0.05
*Trifolium ochroleucon*	2	60.0	*p* < 0.001
*Trifolium pratense*	3	46.8	*p* < 0.001
*Trifolium repens*	2	46.7	*p* < 0.001
*Achillea stricta*	2	55.1	*p* < 0.001
*Alchemilla vulgaris*	3	38.2	*p* < 0.05
*Leucanthemum vulgare*	2	38.8	*p* < 0.01
*Carex pubescens*	2	52.2	*p* < 0.001
*Campanula serata*	2	35.3	*p* < 0.05
*Campanula abietina*	1	41.7	*p* < 0.001
*Cerastium sylvaticum*	1	35.9	*p* < 0.001
*Cruciata glabra*	1	42.9	*p* < 0.001
*Luzula multiflora*	2	48.3	*p* < 0.001
*Hyeracium pilosela*	2	43.5	*p* < 0.01
*Hypericum maculatum*	2	51.7	*p* < 0.001
*Prunela vulgaris*	2	45.2	*p* < 0.01
*Polygala vulgaris*	1	52.4	*p* < 0.001
*Plantago lanceolata*	2	35.4	ns
*Potentila erecta*	2	62.9	*p* < 0.001
*Ranunculas acris*	3	27.9	ns
*Stellaria gramineae*	3	42.0	*p* < 0.05
*Taraxacum officinale*	4	34.8	ns
*Thymus pullegioides*	2	56.3	*p* < 0.001
*Viola declinata*	1	53.8	*p* < 0.001
*Veronica officinalis*	1	52.0	*p* < 0.001
*Veronica chamaedrys*	2	37.0	ns
*Vaccinium myrtillus*	1	61.9	*p* < 0.001

Note: Groups correspond to the four phytosociological clusters identified by cluster analysis. Significance: ns = not significant. Group 1 = *Nardus stricta* (oligotrophic grasslands); Group 2 = *Nardus stricta–Festuca rubra* (transitional communities); Group 3 = Festuca rubra–Nardus stricta co-dominated by *Agrostis capillaris*; Group 4 = *Agrostis capillaris–Festuca rubra* (mesotrophic grasslands).

**Table 5 plants-14-03397-t005:** The influence of fertilization treatments on biodiversity indices in grasslands.

Treatment	Species Richness (S)	Shannon Index (H’)	Evenness (E)	Simpson (D)
T1	35.50 ± 0.50 a	1.54 ± 0.02 e	0.43 ± 0.01 d	0.51 ± 0.01 d
T2	31.25 ± 0.48 b	1.30 ± 0.04 f	0.38 ± 0.01 d	0.42 ± 0.02 d
T3	35.00 ± 0.00 a	1.79 ± 0.05 d	0.50 ± 0.02 c	0.58 ± 0.02 c
T4	37.00 ± 0.00 a	2.59 ± 0.04 a	0.72 ± 0.01 a	0.84 ± 0.01 a
T5	27.50 ± 1.55 c	2.37 ± 0.04 b	0.72 ± 0.01 a	0.84 ± 0.01 a
T6	22.50 ± 0.65 d	2.15 ± 0.01 c	0.69 ± 0.01 ab	0.80 ± 0.00 b
T7	38.00 ± 0.00 a	2.54 ± 0.05 a	0.70 ± 0.01 ab	0.83 ± 0.01 a
T8	28.25 ± 1.11 c	2.49 ± 0.04 a	0.75 ± 0.01 a	0.86 ± 0.01 a
T9	37.00 ± 0.00 a	2.62 ± 0.01 a	0.73 ± 0.00 a	0.85 ± 0.00 a
T10	22.75 ± 0.48 d	2.24 ± 0.02 bc	0.72 ± 0.00 a	0.83 ± 0.00 a
T11	28.00 ± 1.47 c	2.5 ± 0.05 a	0.75 ± 0.016 a	0.86 ± 0.01 a
F test	51.72 (df = 10.33)	156.03 (df = 10.33)	160.32 (df = 10.33)	267.54 (df = 10.33)
*p*-value	*<0.001*	*<0.001*	*<0.001*	*<0.001*

Note: T1—unfertilized control; T2—abandonment; T3—mulching; T4—50 kg N ha^−1^, 50 kg P_2_O_5_ ha^−1^, and 50 kg K_2_O ha^−1^ mineral fertilization; T5—100 kg N ha^−1^, 100 kg P_2_O_5_ ha^−1^, and 100 kg K_2_O ha^−1^ mineral fertilization; T6—150 kg N ha^−1^, 150 kg P_2_O_5_ ha^−1^, and 150 kg K_2_O ha^−1^ mineral fertilization; T7—10 t ha^−1^ cattle manure annually; T8—20 t ha^−1^ cattle manure annually; T9—20 t ha^−1^ cattle manure applied every two years; T10—30 t ha^−1^ cattle manure annually; T11—30 t ha^−1^ cattle manure applied every two years. Note: Values are means ± s.e. (n = 4 replicates per treatment). Different letters within a column indicate significant differences at *p* < 0.05 according to the LSD test.

**Table 6 plants-14-03397-t006:** Modified Braun–Blanquét scale for assessing the abundance–dominance of plant species, based on classes and sub-classes [81].

Class	Coverage Interval (%)	Class Central Value (%)	Sub-Note	Sub-Interval (%)	Central-Adjusted Value of Sub-Interval (%)
5	75–100	87.5	5c	92–100	96
			5b	83–92	87.5
			5a	75–83	79
4	50–75	62.5	4c	67–75	71
			4b	58–67	62.5
			4a	50–58	54
3	25–50	37.5	3c	42–50	46
			3b	33–42	37.5
			3a	25–33	29
2	10–25	17.5	2c	20–25	22.25
			2b	15–20	17.5
			2a	10–15	12.5
1	1–10	5	1c	6–10	8
			1b	4–6	5
			1a	1–4	2.5
+	0.1–1	0.5	-	-	0.5
*r*	0.01–0.1	0.05	-	-	0.05

Note: a–c indicate the sub-note of each class.

## Data Availability

The original contributions presented in this study are included in the article. Further inquiries can be directed to the corresponding authors.

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
