# Peer review of "The Impact of Fertilizer Gradient on High Nature Value Mountain Grassland"

_plants, 2025, doi:10.3390/plants14213397_

Round 1

Reviewer 1 Report

Comments and Suggestions for Authors

The overall quality of the paper is good, but several issues remain, as detailed below:

The language is generally fluent, but numerous spelling errors, inconsistent formatting, and unclear expressions persist. For example, "t ha⁻¹" is sometimes written as "t/ha"; it is recommended to consistently use "t ha⁻¹". The title of Table 5 is "Plant species correlation with the ordination axes (PCoA)", but the content actually presents the results of the Indicator Species Analysis (ISA).

In the Introduction, the sentence “In this sense, European agricultural policies promote agri-environmental schemes and compensatory measures that support biodiversity, yet abandonment and intensification continue to pose significant threats [39,40,8].” lacks a smooth transition to the subsequent sentence “In this context, the objective of this study was to evaluate different management scenarios...”. Consider adding a bridging sentence such as: “To inform more effective management strategies, a clear understanding of how specific practices, such as the type and intensity of fertilization, affect these ecosystems is essential.”

Lines 447-449: Delete this paragraph, as its main idea is already expressed in the Introduction.

In line 454, "luvosoil" and in line 457, "luvosol" – are these the same? Please check and confirm.

Line 457: The meaning of "The soil mountain-meadow" is unclear. Please rephrase.

Lines 465-475: The descriptions of the V1-V11 treatments are unclear in parts, e.g., what is N50P50K50? Please clarify the description of this section. Using a simple diagram to describe the experimental design would improve clarity.

Line 479: P2O5 and K2O should use subscripts for correct chemical formula notation.

Line 482: "4.3. Vegetation survey" should be formatted consistently with other subtitles.

Lines 491-495: This passage is unclear. Please rephrase.

Line 535: In "[89];", remove the semicolon.

Lines 544-545: Is the citation style "after [76]" appropriate?

Lines 541-542: Were tests for normality and homogeneity of variances conducted prior to data analysis?

Line 309, line 353, line 404: Please add numbered subheading labels before these discussion subheadings.

Line 331: Change the citation order from [47, 23, 28, 29] to [23, 28, 29, 47]. Please correct other similar citation sequences accordingly.

Discussion Section: Strengthen the theoretical discussion by explicitly linking the diversity peak under moderate fertilization to the mechanism of the Intermediate Disturbance Hypothesis, rather than just stating it confirms the hypothesis.

Discussion Section: Consolidate the paragraph on plant specificity to avoid redundancy with the Results section and instead synthesize the findings to emphasize their practical application for bio-indication and adaptive management.

Discussion Section: Refine the concluding statement to be more impactful and forward-looking, transforming it from a descriptive conclusion into a persuasive, actionable recommendation for sustainable management.

Author Response

Dear Reviewer, we appreciate your constructing feedback on our work. All your suggestions and comments help us to improve our manuscript. Thank you for all suggestions.

The overall quality of the paper is good, but several issues remain, as detailed below:

The language is generally fluent, but numerous spelling errors, inconsistent formatting, and unclear expressions persist. For example, "t ha⁻¹" is sometimes written as "t/ha"; it is recommended to consistently use "t ha⁻¹". The title of Table 5 is "Plant species correlation with the ordination axes (PCoA)", but the content actually presents the results of the Indicator Species Analysis (ISA).

 We have modified in the text for quantitative expressions and the Title for Table 5.

In the Introduction, the sentence “In this sense, European agricultural policies promote agri-environmental schemes and compensatory measures that support biodiversity, yet abandonment and intensification continue to pose significant threats [39,40,8].” lacks a smooth transition to the subsequent sentence “In this context, the objective of this study was to evaluate different management scenarios...”. Consider adding a bridging sentence such as: “To inform more effective management strategies, a clear understanding of how specific practices, such as the type and intensity of fertilization, affect these ecosystems is essential.”

 I added a linking sentence as per your suggestion.

Lines 447-449: Delete this paragraph, as its main idea is already expressed in the Introduction.

 We have deleted the sentence from Materials and Methods.

In line 454, "luvosoil" and in line 457, "luvosol" – are these the same? Please check and confirm.

 We have checked and inserted the “luvisol” in the entire text.

Line 457: The meaning of "The soil mountain-meadow" is unclear. Please rephrase.

We have rephrased.

Lines 465-475: The descriptions of the V1-V11 treatments are unclear in parts, e.g., what is N50P50K50? Please clarify the description of this section. Using a simple diagram to describe the experimental design would improve clarity.

We have added the explanation for qualitative impact of treatments, based on the quantity and application rate. We have improved the description of the treatments.

Line 479: P2O5 and K2O should use subscripts for correct chemical formula notation.

We have modified in the manuscript text.

Line 482: "4.3. Vegetation survey" should be formatted consistently with other subtitles.

We have modified.

Lines 491-495: This passage is unclear. Please rephrase.

We have updated and rephrased the suggested lines.

Line 535: In "[89];", remove the semicolon.

We have removed.

Lines 544-545: Is the citation style "after [76]" appropriate?

We have updated the reference style.

Lines 541-542: Were tests for normality and homogeneity of variances conducted prior to data analysis?

We have used the MRPP procedure to explore the data, as proposed in the methodology of PC-Ord, due to its performance in the analysis of vegetation groups obtained from floristic surveys. We have analyzed in this way the qualitive aspect of treatments on vegetation cover and assemblage.

Line 309, line 353, line 404: Please add numbered subheading labels before these discussion subheadings.

 We have remediated this omission.

Line 331: Change the citation order from [47, 23, 28, 29] to [23, 28, 29, 47]. Please correct other similar citation sequences accordingly.

 We have checked the entire manuscript.

Discussion Section: Strengthen the theoretical discussion by explicitly linking the diversity peak under moderate fertilization to the mechanism of the Intermediate Disturbance Hypothesis, rather than just stating it confirms the hypothesis.

We have added supplementary explanations.

Discussion Section: Consolidate the paragraph on plant specificity to avoid redundancy with the Results section and instead synthesize the findings to emphasize their practical application for bio-indication and adaptive management.

We have added supplementary information.

Discussion Section: Refine the concluding statement to be more impactful and forward-looking, transforming it from a descriptive conclusion into a persuasive, actionable recommendation for sustainable management.

Thank you for all of your suggestions and comments, they have improved our work.

Kind regards,

The authors.

Reviewer 2 Report

Comments and Suggestions for Authors

Dear Editor and Authors,

After evaluating the manuscript titled “The impact of fertilization type and intensity on the structure and diversity of High Nature Value (HNV) mountain grasslands”, my recommendation is Major Revision.

Overview: The study investigates the effects of fertilization type and intensity (organic vs. mineral) on the floristic composition and diversity of High Nature Value (HNV) grasslands. However, I find the manuscript lacks innovation, presenting limited readership interest and low potential for citation, thus representing a modest scientific contribution. Nevertheless, if the authors significantly improve the introduction, they might convince me otherwise. The manuscript requires a Major Revision to resolve structural inconsistencies, clarify the presentation of methods and results, standardize the nomenclature of experimental variants, and correct table numbering errors. The article should follow the conventional structure: Introduction, Materials and Methods, Results, Discussion, and Conclusions. The Materials and Methods section must precede the Results, ensuring clarity, coherence, and compliance with standard scientific format. Formatting must adhere to Plants (MDPI) guidelines. For example: (i) Each word in the title should begin with a capital letter; (ii) Only the first letter of each surname should be capitalized; (iii) Figure 1 and Tables 4 and 5 titles should not be indented; (iv) Discussion subsections must be numbered consistently with other sections; (v) A period should be placed at the end of the final sentence of the conclusion; (vi) References must have abbreviated journal titles, and article numbers or page ranges should not be italicized, only the journal name and volume should be.

Additional issues found:

Title: Too long; it should have no more than 15 words.

Abstract: Exceeds the journal limit (over 200 words).

Introduction: It is evident that, although extensive, the introduction is insufficient and does not contribute to the understanding of the study. The justification fails to provide a solid basis to convince the reader of the research’s relevance. The research problem is not clearly presented, making it difficult to grasp the importance and significance of the work. In short, the introduction is inadequate for understanding the study. Furthermore, the text is poorly structured, containing only four paragraphs; it is necessary to divide the different topics into more paragraphs. Paragraphs serve precisely to separate topics, which is not being done here. After separating the paragraphs by topic, care should be taken not to repeat the same topic in subsequent paragraphs and to ensure smooth transitions between topics so that the text reads fluently. Finally, the study hypothesis must be clearly stated and aligned with the research objectives.

Results: Before presenting the Results section, the Materials and Methods section should be included. Figure 1 should be improved by increasing the font size of the labels used. The Results section contains excessive repetition of methodological justification. For example, the PCoA description appears in Section 2.2 and again in Section 2.3, where the explained variation percentages and axis meanings are repeated. It is important to emphasize that the Results section should only present the findings. There is too much explanation of how results were obtained and justification of findings; these texts should be moved to the appropriate Methods and Discussion sections. Figures and tables should be presented immediately after the paragraph in which they are cited. There are two Section 2.4s and significant confusion in table and text numbering. There seems to be inconsistency in table numbering in Section 2.4: the ISA analysis is introduced referring to Table 5, but the preceding table listing species correlations is Table 4. The ISA results table following the introduction is again labeled Table 5. Additionally, Table 4 is the first presented after Table 2, indicating a possible missing Table 3 or incorrect numbering of subsequent tables. The Results section also needs to be more concise. There is a large volume of numerical information and unnecessary repetition among methods (Cluster, PCoA, MRPP, and ISA), without prioritizing the most relevant findings. There is also redundancy between the PCoA and MRPP results, which could be integrated into a single analysis. The excessive statistical details (correlation values, significance levels, percentages) compromise readability and should be moved to tables or the Discussion section.

Discussion: The title of the initial subsection of the Discussion is redundant and not very informative, as the entire Discussion addresses this topic. There is an excessive description of results, with little critical interpretation and insufficient exploration of the ecological mechanisms explaining the observed trends. The discussion on climatic variability lacks specific climate data from the study period (2020–2024) that could be directly integrated with the fertilization results, making it seem like an external factor not fully supported by the experiment’s internal data. Broader interpretations relating the findings to practical implications for sustainable management of high nature value grasslands are missing. Finally, the section lacks a critical analysis of the study’s limitations, such as the influence of uncontrolled environmental factors, the duration of observation, or spatial representativeness.

Materials and Methods: The methodology is incomplete and requires more detailed clarification of the proposed procedures. Many basic pieces of information are missing, and as currently presented, a reader would not be able to replicate the experiment. In fact, it is difficult to understand whether the methodology adequately addresses the study objectives. Some of the presented procedures are also questionable. The manuscript lacks essential information for the study to be considered complete and reproducible. Detailed soil data are missing, including initial nutrient concentrations (N, P2O5, K2O), organic matter, cation exchange capacity (CEC), and base saturation, along with a clear description of the timing and method of soil sampling and analysis. Similarly, details on plant management and the harvesting process are insufficient, as the frequency, height, and method of cutting are not specified, nor is it clear whether grazing or plot isolation occurred, information essential to understanding grassland responses under different treatments. Regarding organic fertilization, the dry matter content of cattle manure is not indicated, preventing accurate calculation of nutrient input for each applied dose. Finally, annual or monthly climate data for the entire experimental period (2020–2024) must be included, as the study discusses the influence of climatic variability on the results without presenting the meteorological information needed to support such analysis. There is also inconsistency in the nomenclature of experimental variants: Section 4.2 lists the variants as V1–V11, whereas all other sections refer to them as T1–T11. Section 4.4 reiterates the classification of treatments into input categories (“Zero-input”, “Low-input”, “Medium-input”, “High-input”), which resembles a data interpretation or analysis strategy rather than a standard method description. There is a discrepancy in the number of replicates: Section 4.2 states the experiment was organized in three replicates, but the captions of Figures 1 and 2 indicate R1–R4, implying four replicates.

Conclusions: The conclusion should be rewritten, as it currently resembles a summary of the results. There is excessive repetition of information already discussed, making the text more descriptive than conclusive. Additionally, the conclusion takes a generalizing tone, extrapolating inferences to other environments without clarifying whether such generalizations are supported by the experimental data. Finally, the practical recommendation regarding mowing frequency (“every 4–5 years”) does not appear to be directly supported by the presented results.

References: The references must be formatted according to the journal’s guidelines. Some references are outdated or from sources that were not peer-reviewed; whenever possible, they should be replaced with more recent references from high-impact journals. There are also some self-citations that should be substituted.

Given the above, the manuscript shows many weaknesses, and the recommendation is “Major revision”. I encourage the authors to submit a revised version of the manuscript and, if they agree, I would like to receive responses to all considerations that they disagree with or have not addressed.

Best regards,

Reviewer

Author Response

Response to Reviewer 2

Dear reviewer, thank you for the analysis of our work and the good suggestions and comments. We have tried to modify and improve our manuscript based on your suggestions. Below is a detailed response for each comment and suggestions.

Overview: The study investigates the effects of fertilization type and intensity (organic vs. mineral) on the floristic composition and diversity of High Nature Value (HNV) grasslands. However, I find the manuscript lacks innovation, presenting limited readership interest and low potential for citation, thus representing a modest scientific contribution. Nevertheless, if the authors significantly improve the introduction, they might convince me otherwise. The manuscript requires a Major Revision to resolve structural inconsistencies, clarify the presentation of methods and results, standardize the nomenclature of experimental variants, and correct table numbering errors. The article should follow the conventional structure: Introduction, Materials and Methods, Results, Discussion, and Conclusions. The Materials and Methods section must precede the Results, ensuring clarity, coherence, and compliance with standard scientific format. Formatting must adhere to Plants (MDPI) guidelines. For example: (i) Each word in the title should begin with a capital letter; (ii) Only the first letter of each surname should be capitalized; (iii) Figure 1 and Tables 4 and 5 titles should not be indented; (iv) Discussion subsections must be numbered consistently with other sections; (v) A period should be placed at the end of the final sentence of the conclusion; (vi) References must have abbreviated journal titles, and article numbers or page ranges should not be italicized, only the journal name and volume should be.

 We have modified the text of the manuscript in all the sections as per your suggestions.

We have formatted the text according to MDPI standards, and we have checked and for Plants journal the Materials and Methods are positioned after the Discussion section. We have modified the Names of the authors according to the standards, we have checked all the captions for tables and figures and added numbers to Discussion section. We have added a period after the last conclusion, as suggested. We have checked all the references to ensure that they are formatted according to the standards. We have updated the Treatment description and experimental design.

Additional issues found:

Title: Too long; it should have no more than 15 words.

We have modified the title as suggested.

Abstract: Exceeds the journal limit (over 200 words).

We have modified the Abstract to be under 200 words.

Introduction: It is evident that, although extensive, the introduction is insufficient and does not contribute to the understanding of the study. The justification fails to provide a solid basis to convince the reader of the research’s relevance. The research problem is not clearly presented, making it difficult to grasp the importance and significance of the work. In short, the introduction is inadequate for understanding the study. Furthermore, the text is poorly structured, containing only four paragraphs; it is necessary to divide the different topics into more paragraphs. Paragraphs serve precisely to separate topics, which is not being done here. After separating the paragraphs by topic, care should be taken not to repeat the same topic in subsequent paragraphs and to ensure smooth transitions between topics so that the text reads fluently. Finally, the study hypothesis must be clearly stated and aligned with the research objectives.

We have modified the Introduction, the position of the sentences to show a flow of information – a general description of HNV ecosystems, a general classification and assessment of HNV grasslands, followed by a paragraph related to the changes in management within the last decades and new concepts for the analysis of ecosystem stability. We have analyzed the situation of Romania`s grasslands in the context of population exodus and management changes, respectively different approaches on soil-plant interactions as indicators of community changes.

The last paragraph was modified and updated to present better the aim of the research, the hypothesis of the study, the research questions and the potential use of the results.

Results: Before presenting the Results section, the Materials and Methods section should be included. Figure 1 should be improved by increasing the font size of the labels used. The Results section contains excessive repetition of methodological justification. For example, the PCoA description appears in Section 2.2 and again in Section 2.3, where the explained variation percentages and axis meanings are repeated. It is important to emphasize that the Results section should only present the findings. There is too much explanation of how results were obtained and justification of findings; these texts should be moved to the appropriate Methods and Discussion sections. Figures and tables should be presented immediately after the paragraph in which they are cited. There are two Section 2.4s and significant confusion in table and text numbering. There seems to be inconsistency in table numbering in Section 2.4: the ISA analysis is introduced referring to Table 5, but the preceding table listing species correlations is Table 4. The ISA results table following the introduction is again labeled Table 5. Additionally, Table 4 is the first presented after Table 2, indicating a possible missing Table 3 or incorrect numbering of subsequent tables. The Results section also needs to be more concise. There is a large volume of numerical information and unnecessary repetition among methods (Cluster, PCoA, MRPP, and ISA), without prioritizing the most relevant findings. There is also redundancy between the PCoA and MRPP results, which could be integrated into a single analysis. The excessive statistical details (correlation values, significance levels, percentages) compromise readability and should be moved to tables or the Discussion section.

 The sequence of parts of the manuscript used by us was consistent with the editorial guidelines for authors of the journal Plants (Abstract, Introduction, Results, Discussion, Materials and Methods, and Conclusions).

We have modified the suggested graph, updated the number of sections and removed the unnecessary values from the text. The parts that belong to Methodology were moved to Materials and Methods. Table numbers were updated to correct the omission of number 3 from tables list. 

Discussion: The title of the initial subsection of the Discussion is redundant and not very informative, as the entire Discussion addresses this topic. There is an excessive description of results, with little critical interpretation and insufficient exploration of the ecological mechanisms explaining the observed trends. The discussion on climatic variability lacks specific climate data from the study period (2020–2024) that could be directly integrated with the fertilization results, making it seem like an external factor not fully supported by the experiment’s internal data. Broader interpretations relating the findings to practical implications for sustainable management of high nature value grasslands are missing. Finally, the section lacks a critical analysis of the study’s limitations, such as the influence of uncontrolled environmental factors, the duration of observation, or spatial representativeness.

We have improved the text based on suggested directions. We have added new text to each of Discussion sub-sections, including the limitation of the study.

Materials and Methods: The methodology is incomplete and requires more detailed clarification of the proposed procedures. Many basic pieces of information are missing, and as currently presented, a reader would not be able to replicate the experiment. In fact, it is difficult to understand whether the methodology adequately addresses the study objectives. Some of the presented procedures are also questionable. The manuscript lacks essential information for the study to be considered complete and reproducible. Detailed soil data are missing, including initial nutrient concentrations (N, P2O5, K2O), organic matter, cation exchange capacity (CEC), and base saturation, along with a clear description of the timing and method of soil sampling and analysis. Similarly, details on plant management and the harvesting process are insufficient, as the frequency, height, and method of cutting are not specified, nor is it clear whether grazing or plot isolation occurred, information essential to understanding grassland responses under different treatments. Regarding organic fertilization, the dry matter content of cattle manure is not indicated, preventing accurate calculation of nutrient input for each applied dose. Finally, annual or monthly climate data for the entire experimental period (2020–2024) must be included, as the study discusses the influence of climatic variability on the results without presenting the meteorological information needed to support such analysis. There is also inconsistency in the nomenclature of experimental variants: Section 4.2 lists the variants as V1–V11, whereas all other sections refer to them as T1–T11. Section 4.4 reiterates the classification of treatments into input categories (“Zero-input”, “Low-input”, “Medium-input”, “High-input”), which resembles a data interpretation or analysis strategy rather than a standard method description. There is a discrepancy in the number of replicates: Section 4.2 states the experiment was organized in three replicates, but the captions of Figures 1 and 2 indicate R1–R4, implying four replicates.

 We have updated the Materials and Methods section and completed it with additional information. We have introduced the climate, manure and soil characteristics, and the mowing frequency and corrected the number of replicates to 4. The study did not take into account the soil changes due to the application of fertilizer, but we consider this a good idea for future interdisciplinary studies. Also, we have added a new paragraph of limitations that state this at the end of the Discussion section.

Conclusions: The conclusion should be rewritten, as it currently resembles a summary of the results. There is excessive repetition of information already discussed, making the text more descriptive than conclusive. Additionally, the conclusion takes a generalizing tone, extrapolating inferences to other environments without clarifying whether such generalizations are supported by the experimental data. Finally, the practical recommendation regarding mowing frequency (“every 4–5 years”) does not appear to be directly supported by the presented results.

 We have reorganized and rewritten the Conclusion section to have less and clearer sentences.

References: The references must be formatted according to the journal’s guidelines. Some references are outdated or from sources that were not peer-reviewed; whenever possible, they should be replaced with more recent references from high-impact journals. There are also some self-citations that should be substituted.

 We have updated the format of the reference list, as required by the journal Plants. We have linked our study to relevant references, replaced older references with more recent ones, and removed references by the lead author.

Thank you for all the constructive suggestions and comments. They helped us to improve our manuscript.

Kind regards,

The authors.

Round 2

Reviewer 2 Report

Comments and Suggestions for Authors

Dear authors and editor,

After a new evaluation of the manuscript “The Impact of Fertilizer Gradient on High Nature Value Mountain Grassland”, my recommendation is: Minor revision.

First, I would like to thank the authors for their response letter, which was very clear, and for the revised version of the manuscript, which shows notable improvements. The authors have corrected some content-related, structural, and formatting issues. However, certain weaknesses remain that prevent me from recommending the manuscript for acceptance at this stage. My considerations are as follows:

(i) Unaddressed comments:

Despite the authors’ efforts to respond to all the reviewers’ comments, two major issues have not been properly implemented in the revised version: It was requested that the paper follow the conventional structure (Introduction, Materials and Methods, Results, Discussion, and Conclusions), emphasizing that the Materials and Methods section should precede Results, in accordance with standard scientific format. Even if the journal allows flexibility, as mentioned by the authors, the conventional and logical order should follow what was requested in the first review. The inclusion of detailed soil data in the Materials and Methods section, such as initial nutrient concentrations, organic matter (OM), cation exchange capacity (CEC), and base saturation, has still not been provided. Therefore, the study remains incomplete and not fully reproducible.

(ii) Partially Addressed Comments:

Some suggestions were implemented, but the solutions presented in the revised manuscript did not fully resolve the issues raised in my first review or introduced new ambiguities. In the first review, I noted that the Results section contained excessive repetition of methodological information and that redundancy between the PCoA and MRPP results could be integrated. The authors stated that they had removed unnecessary values and moved some parts to the Methodology section. However, in the new version of the manuscript, sections 2.2 (PCoA) and 2.3 (MRPP) still show overlapping information, where the main PCoA results (e.g., variation explained by Axis 1 and Axis 2) are presented in Table 1 and then described again in detail in section 2.3, when illustrating species distribution (Axis 1: 87.5%; Axis 2: 11.1%). The separation of the analyses into distinct sections while retaining repeated explanations of the axis meanings indicates that the integration suggested in the first review has not been fully implemented. I previously criticized the inclusion of treatment classification into input categories (“Zero-input”, “Low-input”, “Medium-input”, “High-input”) within the Materials and Methods section, as this approach resembled a data interpretation or analysis strategy rather than a standard methodological description. In the revised version, the authors retained and even expanded this classification in section 4.2 (Experimental Design), stating that it would be used to show a qualitative effect and quantify the fertilizer effect. Although the rationale is understandable, maintaining this classification within Experimental Design in Materials and Methods, rather than addressing it as an analytical variable in the Data Analysis section, still preserves the criticized overlap between methodological description and data interpretation.

(iii) New Critical Points in the Manuscript:

The new version of the manuscript presents additional critical issues or editorial oversights that affect its clarity. The explanatory note for Table 5, which presents biodiversity indices, contains truncated and repetitive text at the end, possibly due to a copy-paste error from another figure or table caption. Table 3 in the revised version lists the correlation between species and PCoA ordination axes. Although the table numbering (1, 2, 3, 4, 5) appears correct, the descriptive text in section 2.3 (MRPP), immediately before Table 3, introduces it with the sentence: “To further explore the contribution of individual species to the separation of phytocenoses highlighted in the PCoA analysis, the correlations between species abundance and the two ordination axes are presented in Table 3”. While this is logically correct, Table 3 actually refers to the PCoA correlations (not MRPP, which corresponds to Table 2). The placement of Table 3, immediately after the MRPP description in section 2.3 (titled The analysis of community composition through Multi-Response Permutation Procedure (MRPP)), may confuse readers regarding which analysis the data belong to (even though the table is related to PCoA). Although the authors changed the title of the first subsection of the Discussion from “The evolution of phytocoenoses in different management scenarios” (version 1) to “3.1. The impact of different management scenarios on plant community” (version 2), I had previously considered the earlier title redundant for describing the general theme of the entire Discussion. The new title (3.1) remains broadly descriptive of the full scope of the Discussion, maintaining the underlying issue that it does not clearly differentiate this subsection from the others (e.g., section 3.3 Structural and diversity changes along input gradients). In section 4.1 of version 2, while describing the soil, there is a citation referring to (Stănilă and Dumitru, 2016), which also appears in the reference list. However, in line 524 of version 2, the citation is inserted immediately after the mention of “luvisol”, which represents a slight inaccuracy in citation placement relative to text flow. The text within Figures 1, 2, and 3 is unreadable due to very small font size and low resolution. The authors should increase the font size for legibility.

Given the above, some weaknesses remain, and my recommendation is “Minor revision”.

I encourage the authors to submit a new version of the manuscript and, if they disagree with any of my comments, to provide specific responses addressing those points.

Best regards,

Reviewer

Author Response

Response to Reviewer

My considerations are as follows:

(i) Unaddressed comments:

Despite the authors’ efforts to respond to all the reviewers’ comments, two major issues have not been properly implemented in the revised version: It was requested that the paper follow the conventional structure (Introduction, Materials and Methods, Results, Discussion, and Conclusions), emphasizing that the Materials and Methods section should precede Results, in accordance with standard scientific format. Even if the journal allows flexibility, as mentioned by the authors, the conventional and logical order should follow what was requested in the first review.

On the Plants journal website (https://www.mdpi.com/journal/plants/instructions), in the Manuscript Preparation subchapter of Instructions for Authors, the following sequence of parts of the manuscript is presented: Abstract, Introduction, Results, Discussion, Materials and Methods, Conclusions, without any specification of modification of the structure by the authors. We have respected this sequence and we hope that you agree with it, which is also found in the articles published in the volumes of the journal.

The inclusion of detailed soil data in the Materials and Methods section, such as initial nutrient concentrations, organic matter (OM), cation exchange capacity (CEC), and base saturation, has still not been provided. Therefore, the study remains incomplete and not fully reproducible.

We have updated the soil characteristics by introducing new data.

(ii) Partially Addressed Comments:

Some suggestions were implemented, but the solutions presented in the revised manuscript did not fully resolve the issues raised in my first review or introduced new ambiguities. In the first review, I noted that the Results section contained excessive repetition of methodological information and that redundancy between the PCoA and MRPP results could be integrated. The authors stated that they had removed unnecessary values and moved some parts to the Methodology section. However, in the new version of the manuscript, sections 2.2 (PCoA) and 2.3 (MRPP) still show overlapping information, where the main PCoA results (e.g., variation explained by Axis 1 and Axis 2) are presented in Table 1 and then described again in detail in section 2.3, when illustrating species distribution (Axis 1: 87.5%; Axis 2: 11.1%). The separation of the analyses into distinct sections while retaining repeated explanations of the axis meanings indicates that the integration suggested in the first review has not been fully implemented.

 We have removed the redundant information regarding the Axis importance and left it only Table 1.

I previously criticized the inclusion of treatment classification into input categories (“Zero-input”, “Low-input”, “Medium-input”, “High-input”) within the Materials and Methods section, as this approach resembled a data interpretation or analysis strategy rather than a standard methodological description. In the revised version, the authors retained and even expanded this classification in section 4.2 (Experimental Design), stating that it would be used to show a qualitative effect and quantify the fertilizer effect. Although the rationale is understandable, maintaining this classification within Experimental Design in Materials and Methods, rather than addressing it as an analytical variable in the Data Analysis section, still preserves the criticized overlap between methodological description and data interpretation.

 We have moved the information regarding Input categories in Data Analysis section, where is more meaningful.

(iii) New Critical Points in the Manuscript:

The new version of the manuscript presents additional critical issues or editorial oversights that affect its clarity. The explanatory note for Table 5, which presents biodiversity indices, contains truncated and repetitive text at the end, possibly due to a copy-paste error from another figure or table caption.

We have removed the repetitive text.

Table 3 in the revised version lists the correlation between species and PCoA ordination axes. Although the table numbering (1, 2, 3, 4, 5) appears correct, the descriptive text in section 2.3 (MRPP), immediately before Table 3, introduces it with the sentence: “To further explore the contribution of individual species to the separation of phytocenoses highlighted in the PCoA analysis, the correlations between species abundance and the two ordination axes are presented in Table 3”. While this is logically correct, Table 3 actually refers to the PCoA correlations (not MRPP, which corresponds to Table 2). The placement of Table 3, immediately after the MRPP description in section 2.3 (titled The analysis of community composition through Multi-Response Permutation Procedure (MRPP)), may confuse readers regarding which analysis the data belong to (even though the table is related to PCoA).

 We have modified the title of section 2.3. to be more informative and the title of Table 3 to be relevant for this section. Now it explains better the results. Section 2.2. is focused on grassland communities, their projection on PCoA and the MRPP between community types, while Section 2.3. is focused on species reaction and projection on PCoA due to the applied treatments.

Although the authors changed the title of the first subsection of the Discussion from “The evolution of phytocoenoses in different management scenarios” (version 1) to “3.1. The impact of different management scenarios on plant community” (version 2), I had previously considered the earlier title redundant for describing the general theme of the entire Discussion. The new title (3.1) remains broadly descriptive of the full scope of the Discussion, maintaining the underlying issue that it does not clearly differentiate this subsection from the others (e.g., section 3.3 Structural and diversity changes along input gradients).

We have changed the title of Section 3.1 to: The changes of different management scenarios in the assemblage of grassland communities. We have modified the title of Section 3.2. to: Species structure and diversity under the input gradients.

In section 4.1 of version 2, while describing the soil, there is a citation referring to (Stănilă and Dumitru, 2016), which also appears in the reference list. However, in line 524 of version 2, the citation is inserted immediately after the mention of “luvisol”, which represents a slight inaccuracy in citation placement relative to text flow.

 We have moved the reference at the end of the sentence.

The text within Figures 1, 2, and 3 is unreadable due to very small font size and low resolution. The authors should increase the font size for legibility.

The format of the figures (Font, colors, abbreviations) is the one resulting from the software used to fit the figures into the page format. We took into account the clarity of the figures' expression.

Thank you for all the constructive suggestions and comments. They helped us to improve our manuscript.

Best regards,

Autors
